# New Bithiophene Extended IDIC-Based Non-Fullerene Acceptors and Organic Photovoltaics Thereof

**DOI:** 10.3390/molecules27031113

**Published:** 2022-02-07

**Authors:** Yeong Heon Jeong, Jae Min Jeon, Jun Young Kim, Yun-Hi Kim

**Affiliations:** 1Department of Chemistry and RINS, Gyeongsang National University, Jinju 660-701, Korea; dudgjs02@naver.com; 2Department of Semiconductor Engineering, Gyeongsang National University, Jinju 660-701, Korea; jmjeon95@gnu.ac.kr

**Keywords:** non-fullerene acceptors, IDIC, organic photovoltaic

## Abstract

We developed new bithiophene extended electron acceptors based on *m*-alkoxythenyl-substituted IDIC with three different end groups, named as IDT-BT-IC, IDT-BT-IC4F, and IDT-BT-IC4Cl, respectively. The ultraviolet absorption maximum was redshifted and the bandgap was decreased as the strong electron accepting ability of the end group increased. A differential scanning calorimetry thermogram analysis revealed that all the new acceptors have a crystalline character. Using these acceptors and a bulk heterojunction structure using PBDB-T, inverted organic photovoltaic (OPV) devices were fabricated, and their performance was analyzed. Due to the red shift of the electron acceptors, the OPV active layer particularly, which was derived from IDT-BT-IC4F, exhibited increased absorption at long wavelengths over 800 nm. The OPV prepared using IDT-BT-IC exhibited a short-circuit current density (J_sc_) of 2.30 mA/cm^2^, an open-circuit voltage (V_oc_) of 0.95 V, a fill factor (FF) of 45%, and a photocurrent efficiency (PCE) of 1.00%. Using IDT-BT-IC4F, the corresponding OPV device showed J_sc_ = 8.31 mA/cm^2^, V_oc_ = 0.86 V, FF = 47%, and PCE = 3.37%. The IDT-BT-IC4Cl-derived OPV had J_sc_ = 3.00 mA/cm^2^, V_oc_ = 0.89 V, FF = 29%, and PCE = 0.76%. When IDT-BT-IC4F was used as the electron acceptor, the highest J_sc_ and PCE values were achieved. The results show that the low average roughness (0.263 nm) of the active layer improves the extraction of electrons.

## 1. Introduction

The development of photovoltaic technologies has advanced rapidly with the increasing demand for renewable energy sources to solve environmental pollution. Among the available solar cell technologies, organic photovoltaics (OPVs) stand out due to their easy processability, low cost, light weight, and flexibility [1,2,3,4,5,6].

In general, OPVs are prepared by constructing a bulk heterojunction (BHJ), which is an active layer comprising a blend of an electron donor (D) and electron acceptor (A), via a solution process, which provides large-area OPVs using the roll-to-roll process [4,5,6,7,8].

In contrast with the intensive research that has been conducted to obtain donor materials, the development of acceptor materials has lagged behind because fullerene derivatives are commonly used as electron acceptors. Fullerene derivatives are characterized by a fully conjugated sp^2^-hybridized structure, which provides excellent electron transport and acceptor capacity and promotes electron delocalization. Despite having these desirable properties, the three-dimensional structure of fullerenes limits their practical application. Thus, these compounds suffer from weak light absorption, difficulty in energy state control and backbone control, and photochemical and morphological instability [1,2,3,9]. Consequently, the development of non-fullerene acceptors (NFAs) has been actively pursued. Representative non-fullerene acceptors are BTP-eC9 [10], BTP-4F-PC6 [11], Y6 [12], and IDIC [13]. With high absorption coefficient due to their planar two-dimensional structure, NFAs can be easily tuned to achieve energy compatibility and optical complementarity with various donor polymers, affording a wide solar spectrum coverage. In addition, their easy synthesis and purification processes can reduce the production cost compared with fullerenes [1,2,3,6,14,15,16,17,18]. NFAs based on indaceno[1,2-b:5,6-b′]dithiophene (IDT) with electron-rich ring-fused linear ladder structures [19], such as IDIC or ITIC derivatives, are ring-fused materials having a push–pull A–D–A structure [20,21]. These A–D–A-type IDIC or ITIC derivatives have a high power conversion efficiency of around 14%.

In this work, we designed A–D′–D–D′–A-structured IDIC derivatives with extended bithiophenes to increase the LUMO energy level for high open-circuit voltage (V_oc_). The physical properties of bithiophene extended ITIC derivatives were also compared with those of IDIC derivatives. Moreover, the performance of OPV devices fabricated using these derivatives was studied.

## 2. Results and Discussion

### 2.1. NFA Characteristics

Figure 1 depicts the synthesis of the materials, which involved various organic reactions such as formylation, Stille coupling reaction, nucleophilic substitution, and Knoevenagel condensation. The structures of intermediate materials and the final compounds were characterized by ^1^H nuclear magnetic resonance (NMR), ^13^C NMR, and mass spectroscopies. (Appendix A). The details of their synthesis and characterization are described in the Appendix A. As a donor material, poly[(2,6-(4,8-bis(5-(2-ethylhexyl)thiophene-2-yl)-benzo[1,2-b:4,5-b′]dithiophene))-alt-(5,5-(1′,3′-di-2-thienyl-5′,7′-bis(2-ethylhexyl) benzo[1′,2′-c:4′,5′-c′]dithiophene-4,8-dionz) (PBDB-T) was used.

The optical properties of the new NFAs were investigated by ultraviolet–visible (UV–vis) spectroscopy in chloroform solution at a certain concentration (1 × 10^−5^ M) and in film state (Appendix A). In solution, IDT-BT-IC, IDT-BT-IC4F, and IDT-BT-IC4Cl showed absorption maxima at 682, 710, and 726 nm, respectively, which were redshifted with increasing the electronegativity of the end group. Thus, the UV absorption maximum of IDT-BT-IC was redshifted by almost 50 nm compared with ITIC-OEh without the bithiophene unit [22]. This suggests that the introduction of the bithiophene unit extended the conjugation in the molecule. A redshift of 50–70 nm was observed for all the compounds in the film state. Meanwhile, the optical bandgap was gradually decreased as the electronegativity of the end group increased. The HOMO and LUMO energies of the new NFAs were determined by cyclic voltammetry and optical bandgap measurements (Appendix A). Similar values of −5.25, −5.25, and −5.26 eV were obtained for the HOMO levels of IDT-BT-IC, IDT-BT-IC4F, and IDT-BT-IC4Cl, respectively, whereas the LUMO energy level of IDT-BT-IC, IDT-BT-IC4F, and IDT-BT-IC4Cl was −3.82, −3.86, and −3.88 eV, respectively. The optical and electrochemical properties are summarized in Table 1. The thermal stability of IDT-BT-IC, IDT-BT-IC4F, and IDT-BT-IC4Cl was examined by thermogravimetry analysis and differential scanning calorimetry. For all NFAs, the decomposition temperature at 5% weight loss was observed at above 335 °C. The ID-BT-based new NFAs showed a crystalline nature with a melting transition at 267–276 °C and a crystallization transition at 294–229 °C. (Appendix A).

### 2.2. OPV Characteristics

Figure 1 shows the characteristics of a series of OPV devices fabricated using IDT-BT-IC, IDT-BT-IC4F, and IDT-BT-IC4Cl by spin coating. The absorption spectra of the thin films depicted in Figure 1a shows that all NFAs produced a maximum absorption peak at about 620 nm with a shoulder from ca. 700 to 800 nm and a less intense peak at about 580 nm. These peaks are weaker in the case of IDT-BT-IC4F. Figure 1b shows the changes in the current density with voltage under light irradiation at 100 mW/cm^2^. Detailed parameter values of Figure 1b are shown in Table 1. IDT-BT-IC afforded the lowest short-circuit current density (J_sc_) of 2.30 mA/cm^2^ among the NFAs. The corresponding values for IDT-BT-IC4Cl and IDT-BT-IC4F were 3.00 and 8.31 mA/cm^2^, respectively, which indicates that the current density increased when adding a halogen element.

The V_oc_ of IDT-BT-IC without halogen elements was 0.95 V, which was 0.06~0.09 V higher than that of the other two NFAs (0.86 V for IDT-BT-IC4F and 0.89 V for IDT-BT-IC4Cl). When designing the structure, high V_oc_ was expected due to the improvement of the LUMO level, but it seems that the maximum V_oc_ did not come out due to changes in other factors. V_loss_ is one of the most important factors limiting the PCE of the OPVs. In particular, the contribution from non-radiative V_loss_ should be reduced, which is mainly affected by the energetic interactions between the donor polymer and acceptors at the interfaces. Thus, the interfacial and morphological properties (i.e., domain size/purity, aggregation, and molecular orientation) of the BHJ blend should be optimized to decrease the non-radiative V_loss_ [23]. The fill factor (FF), which is the ratio of the actual maximum obtainable power to the product of I_sc_ and V_oc_, is greatly affected by internal resistance, with large shunt resistance (R_sh_) and small series resistance (R_s_) improving the FF [24,25]. The R_s_ and R_sh_ can be obtained from the J–V graph under light irradiation at 100 mW/cm^2^. R_s_ and R_sh_ are the inverses of the slopes at V_oc_ and J_sc_. These conditions were not satisfied by IDT-BT-IC and IDT-BT-IC4F; IDT-BT-IC had high R_s_ and R_sh_ values of 56 and 830 Ω cm^2^, respectively, and IDT-BT-IC4F showed low R_sh_ and R_s_ values of 337 and 18 Ω cm^2^, respectively, affording similar FF values of 45% and 47%. Therefore, satisfying only one of the conditions (high R_sh_ or low R_s_) did not give a high FF [26]. Meanwhile, IDT-BT-IC4Cl showed a lower FF of about 29% due to its low R_sh_ (361 Ω cm^2^) and high R_s_ (139 Ω cm^2^). As shown in Table 1, IDT-BT-IC4Cl had the lowest photocurrent efficiency (PCE) of 0.76% due to its low current density (3.00 mA/cm^2^) and FF (29%). Furthermore, IDT-BT-IC showed a PCE of 1.00% owing to its low current density (2.30 mA/cm^2^) and high FF (45%), and IDT-BT-IC4F showed the highest PCE of 3.37% due to its high current density (8.31 mA/cm^2^) and FF (47%). Figure 1c shows the J–V plot measured in the dark state, which is a semi-log graph in which the y axis is expressed on a log scale to observe the characteristics of the current density. In the reverse bias voltage region below 0 V, IDT-BT-IC exhibited lower current density than the other two devices, which is indicative of lower leakage current. A low leakage current is closely related to R_sh_. Thus, the lower leakage current of IDT-BT-IC can be attributed to its higher R_sh_ compared with that of IDT-BT-IC4F and IDT-BT-IC4Cl. We extracted the series resistance in the 4–5 V range of dark I–V in Figure 1c, and expressed it as R_s dark_. The value of R_s_ dark is 1.35, 1.31, 40.27 Ω cm^2^ for IDT-BT-IC, IDT-BT-IC4F and IDT-BT-IC4Cl respectively. IDT-BT-IC4Cl has high series resistance at dark current, It shows a similar trend to R_s_ in the J–V graph under light irradiation at 100 mW/cm^2^. Figure 1d shows the changes in the external quantum efficiency (EQE) with the wavelength in the range from 350 to 800 nm. For all the wavelength range, IDT-BT-IC4F showed the highest EQE. EQE represents the generation of an electron–hole pair upon absorption of light in the active layer, followed by charge separation and transfer to both electrodes, generating a current. In other words, light absorption is crucial, but it is also essential to allow the generation of electron–hole pairs as an electric current without recombination. Therefore the absorption of the photoactive layer and the EQE of the OPV device do not entirely match. This is because the OPV device is not only the photoactive layer but also may be caused by the material of the charge extraction and the roughness of the interface layer so that the absorption and EQE graphs do not match [27].

As shown in Figure 1a, IDT-BT-IC4F had the lowest absorption rate; however, its EQE value was significantly higher than that of the other two devices. This indicates that charge recombination is prevented after electrons and holes are separated, which means that the generation of current is enhanced in IDT-BT-IC4F.

Figure 2 shows atomic force microscopy (AFM) images of various photoactive layers. The size of the AFM image was 5 μm × 5 μm. According to these images, the average roughness (R_a_) of the surface was 1.649 nm for IDT-BT-IC, 0.263 nm for IDT-BT-IC4F, and 0.430 nm for IDT-BT-IC4Cl. The high R_a_ of IDT-BT-IC limits the movement of electron–hole pairs, resulting in the lowest current density value of 2.30 mA/cm^2^. Similarly, its leakage current was low because it prevents the movement of charges by the electric field at reverse bias voltage. In contrast, the halogen-containing photoactive layers exhibited low surface roughness. Among these two devices, IDT-BT-IC4Cl showed many pinholes and white spots because the arrangement between particles was not dense, which results in a low current density of 3.00 mA/cm^2^. IDT-BT-IC4F had few pinholes or white spots, which provides a flat surface for a smooth charge transfer, thereby improving the interface performance with MoO_3_ to achieve the highest current density of 8.31 mA/cm^2^. As a result, although the absorption of PBDB-T:IDT-BT-IC4F is the lowest at 600 nm, the reason why the EQE is the highest is because the morphological properties are greatly improved compared to other PBDB-T:NFAs.

## 3. Materials and Methods

### 3.1. Synthesis of the NFAs

#### 3.1.1. Synthesis of 4,4,9,9-Tetrakis(3-((2-ethylhexyl)oxy)phenyl)-3,8-dioctyl-4,9-dihydro-s-indaceno[1,2-b:5,6-b′]dithiophene (**6**)

In a three-neck round bottom flask (DAIHAN Scientific Co, Ltd., Wonju, Korea), to a stirring solution of 1-bromo-3-((2-ethylhexyl)oxy)benzene (9.08 g, 31.85 mmol) in dried THF (70 mL) was added dropwise a 2.5 M solution of *n*-BuLi in *n*-hexane (12.74 mL) under N_2_ condition at −78 °C (Merk KGaA, Darmstadt, Germany). After 1.5 h at this temperature, the solution temperature was raised slowly to −65 °C. Then, a solution of compound **5** (75 mL, 6.37 mmol) in THF was added dropwise into the solution at −65 °C. After slowly warming up to ambient temperature, the mixture was kept for 8 h, followed by addition of water, and the mixture was extracted with ethyl acetate and dried over MgSO_4_. After evaporating the solvent, the hydroxy intermediate was dissolved with glacial acetic acid (300 mL), and the solution of the intermediate was then refluxed for 3 h under N_2_ condition. After cooling to ambient temperature, water was added and the mixture was extracted with dichloromethane, washed with water, dried over MgSO_4_, and concentrated. The residue was purified by column chromatography (Merk KGaA, Darmstadt, Germany) using methylene chloride: *n*-hexane (*v*:*v* = 1:3) as an eluent to obtain the product as a white solid (2.5 g, 30%). ^1^H NMR (300 MHz, CD_2_Cl_2_, δ): 7.41 (s, 2H), 7.15–7.20 (t, 4H), 6.93 (s, 2H), 6.88–6.90 (t, 4H), 6.79–6.83 (q, 8H), 3.78–3.80 (d, *J* = 5.82 Hz, 8H), 2.32–2.36 (t, 4H), 1.65–1.73 (m, 4H), 1.18–1.54 (m, 56H), 0.89–0.94 (m, 30H). ^13^C NMR (500 MHz, CDCl_3_, δ): 159.08, 155.46, 153.34, 146.87, 145.04, 144.06, 142.88, 141.83, 139.16, 138.33, 136.39, 135.52, 128.93, 121.92, 121.10, 120.86, 115.95, 115.68, 112.51, 70.47, 63.58, 39.32, 33.41, 31.90, 30.48, 29.46, 29.37, 29.28, 29.21, 29.05, 28.93, 23.79, 23.05, 22.68, 14.09, 11.07. HR-MS (FAB) *m*/*z* C_88_H_123_O_4_S_2_ Calcd: 1306.8785, Found: 1307.8843.

#### 3.1.2. Synthesis of 2,7-Dibromo-4,4,9,9-tetrakis(3-((2-ethylhexyl)oxy)phenyl)-3,8-dioctyl-4,9-dihydro-s-indaceno[1,2-b:5,6-b′]dithiophene (**7**)

To a stirring solution of compound **6** (2.21 g, 1.69 mmol) in dried CHCl_3_ (80 mL), *N*-bromosuccinimide (0.66 g, 3.72 mmol) was added three times at ambient temperature, and the mixture was kept for 8 h in the dark. After pouring water, the mixture was extracted with CHCl_3_ and dried over MgSO_4_. After evaporating the solvent, the mixture was purified by column chromatography using silica gel (Merk KGaA, Darmstadt, Germany) and methylene chloride:hexane (*v*:*v* = 1:3) as the eluent. Then, the compound was purified by recrystallization from methylene chloride (1.8 g, 72.7%). ^1^H NMR (300 MHz, CD_2_Cl_2_, δ): 7.37 (s, 2H), 7.15–7.21 (t, 4H), 6.88 (s, 4H), 6.77–6.83 (t, 8H), 3.78–3.80 (d, 8H), 2.31–2.37 (m, 4H), 1.65–1.73 (m, 4H), 1.10–1.49 (m, 56H), 0.89–0.94 (t, 30H). ^13^C NMR (500 MHz, CDCl_3_, δ): 159.20, 155.06, 152.14, 143.48, 140.45, 138.71, 135.34, 129.14, 120.75, 115.77, 112.57, 111.20, 70.62, 64.35, 39.33, 31.97, 30.48, 29.75, 29.28, 29.20, 29.11, 29.07, 28.37, 28.27, 23.81, 23.08, 22.72, 14.13, 11.11. MS (FAB) *m*/*z* Calcd: 1462.6995, Found: 1466.

#### 3.1.3. Synthesis of 5′,5‴-(4,4,9,9-Tetrakis(3-((2-ethylhexyl)oxy)phenyl)-3,8-dioctyl-4,9-dihydro-s-indaceno[1,2-b:5,6-b′]dithiophene-2,7-diyl)bis(3-octyl-[2,2′-bithiophene]-5-carbaldehyde) (**8**)

Compounds **3** (1.76 g, 1.2 mmol) and **7** (1.79 g, 3.0 mmol) were dissolved in dried toluene (60 mL). After bubbling the solution with N_2_ gas for 30 min, Pd(PPh_3_)_4_ (0.07 g, 0.06 mmol) was added and refluxed for 12 h. After the mixture was kept for 8 h, water was poured and the mixture was extracted with CHCl_3_. After evaporating the solvent, the mixture was purified by silica gel column chromatography using methylene chloride:hexane (*v*:*v* = 2:1) as the eluent. Then, the compound was purified by recrystallization from methylene chloride (1.1 g, 48%). ^1^H NMR (300 MHz, CD_2_Cl_2_, δ): 9.84 (s, 2H), 7.64 (s, 2H), 7.45 (s, 2H), 7.29–7.30 (d, 2H), 7.18–7.24 (m, 6H), 6.97 (s, 4H), 6.82–6.89 (t, 8H), 3.80–3.82 (d, *J* = 5.79 Hz, 8H), 2.83–2.89 (t, 4H), 2.52–2.56 (m, 4H), 1.67–1.77 (m, 8H), 1.07–1.53 (m, 80H), 0.85–0.94 (m, 36H), 0.66–0.71 (m, 4H). ^13^C NMR (500 MHz, CDCl3, δ): 182.49, 159.25, 156.24, 155.30, 143.50, 141.25, 140.76, 140.12, 140.06, 139.43, 139.09, 136.88, 135.35, 133.99, 133.04, 129.15, 127.78, 125.33, 120.91, 115.93, 112.52, 70.65, 70.62, 64.04, 39.33, 31.98, 31.86, 30.47, 30.26, 29.93, 29.49, 29.44, 29.32, 29.27, 29.23, 29.09, 29.04, 23.79, 23.07, 22.71, 22.66, 14.10, 11.10, 11.07. MS (MALDI-TOF) *m*/*z* Calcd: 1915.0696, Found: 1915.1112.

#### 3.1.4. Synthesis of 2,2′-((2Z,2′Z)-(((4,4,9,9-Tetrakis(3-((2-ethylhexyl)oxy)phenyl)-3,8-dioctyl-4,9-dihydro-s-indaceno[1,2-b:5,6-b′]dithiophene-2,7-diyl)bis(3-octyl-[2,2′-bithiophene]-5′,5-diyl))bis(methaneylylidene))bis(3-oxo-2,3-dihydro-1H-indene-2,1-diylidene))dimalononitrile (IDT-BT-IC)

In a three-neck round bottom flask, to a stirring solution of compound **8** (0.2 g, 0.10 mmol) and 2-(3-oxo-2,3-dihydro-1H-inden-1-ylidene)malononitrile (0.1 g, 0.52 mmol) in dried CHCl_3_ (30 mL), pyridine (1 mL) was added (Zhejiang Boom King Industrial Co., Ltd., Zhejiang, China). After the mixture was refluxed for 8 h, water was poured and the mixture was extracted with CHCl_3_ and dried over MgSO_4_. After evaporating the solvent, the mixture was purified by silica gel column chromatography using methylene chloride:hexane (*v*:*v* = 3:1) as the eluent. Then, recrystallization from CHCl_3_ afforded IDT-BT-IC as a black solid (0.15 g, 63.3%). ^1^H NMR (300 MHz, CD_2_Cl_2_, δ): 8.82 (s, 2H), 8.70–8.73 (q, 2H), 7.94–7.96 (m, 2H), 7.79–7.83 (m, 4), 7.74 (s, 2H), 7.52–7.53 (d, *J* = 4.04 Hz, 2H), 7.48 (s, 2H), 7.27–7.28 (d, *J* = 4.02 Hz, 2H), 7.20–7.22 (t, 4H), 6.99 (s, 4H), 6.83–6.90 (m, 8H), 3.81–3.83 (d, *J* = 5.91 Hz, 8H), 2.87–2.93 (t, 4H), 2.61–2.66 (m, 4H), 1.68–1.80 (m, 8H), 1.10–1.53 (m, 80H), 0.84–0.95 (m, 36H), 0.69–0.73 (m, 4H). ^13^C NMR (500 MHz, CDCl_3_, δ): 159.20, 143.39, 140.04, 136.92, 135.08, 134.43, 129.20, 125.28, 123.73, 120.89, 115.96, 112.53, 70.67, 70.64, 39.32, 31.98, 31.86, 30.46, 30.00, 29.63, 29.44, 29.36, 29.31, 29.24, 29.09, 29.05, 23.78, 23.06, 22.72, 22.66, 14.11, 11.11, 11.08. MS (MALDI-TOF) *m*/*z* Calcd: 2267.1445, Found: 2268.5410.

#### 3.1.5. Synthesis of 2,2′-((2Z,2′Z)-(((4,4,9,9-Tetrakis(3-((2-ethylhexyl)oxy)phenyl)-3,8-dioctyl-4,9-dihydro-s-indaceno[1,2-b:5,6-b′]dithiophene-2,7-diyl)bis(3-octyl-[2,2′-bithiophene]-5′,5-diyl))bis(methaneylylidene))bis(5,6-difluoro-3-oxo-2,3-dihydro-1H-indene-2,1-diylidene)) dimalononitrile (IDT-BT-IC4F)

In a three-neck round bottom flask, pyridine (1 mL) was added to a stirring solution of compound **8** (0.3 g, 0.15 mmol) and 2-(5,6-difluoro-3-oxo-2,3-dihydro-1H-inden-1-ylidene)malononitrile (0.18 g, 0.78 mmol) in dried CHCl_3_ (30 mL), and the mixture was refluxed for 8 h. Water was poured, and the mixture was extracted with CHCl_3_ and dried over MgSO_4_. After evaporating the solvent, the mixture was purified by column chromatography using silica gel and methylene chloride:hexane (*v*:*v* = 3:1) as the eluent. Then, the compound was recrystallized from CHCl_3_, affording IDT-BT-IC4F as a black solid (0.25 g, 68.2%). ^1^H NMR (300 MHz, CD_2_Cl_2_, δ): 8.80 (s 2H), 8.53–8.59 (q, 2H), 7.70–7.75 (t, 4H), 7.53–7.55 (d, *J* = 4.06 Hz, 2H), 7.48 (s, 2H), 7.28–7.29 (d, *J* = 4.05 Hz, 2H), 7.20–7.22 (t, 4H), 7.00 (s, 4H), 6.84–6.90 (m, 8H), 3.81–3.83 (d, *J* = 5.89 Hz, 8H), 2.88–2.94 (t, 4H), 2.61–2.67 (m, 4H), 1.68–1.80 (m, 8H), 1.09–1.54 (m, 80H), 0.85–0.95 (m, 36H), 0.70–0.72 (m, 4H). ^13^C NMR (500 MHz, C_2_D_2_Cl_4_, δ): 185.69, 159.48, 158.70, 156.65, 155.98, 149.94, 148.52, 134.37, 141.73, 141.47, 141.41, 137.87, 137.44, 135.49, 134.38, 134.03, 133.27, 129.56, 129.35, 125.95, 121.65, 121.07, 116.24, 116.16, 114.51, 114.32, 113.23, 71.27, 69.70, 64.35, 39.49, 32.00, 31.92, 30.70, 30.07, 30.01, 29.66, 29.54, 29.48, 29.38, 29.27, 29.22, 29.19, 29.05, 24.10, 23.12, 22.77, 22.72, 14.19, 11.25. MS (MALDI-TOF) *m*/*z* Calcd: 2339.1068, Found: 2339.1575.

#### 3.1.6. Synthesis of 2,2′-((2Z,2′Z)-(((4,4,9,9-Tetrakis(3-((2-ethylhexyl)oxy)phenyl)-3,8-dioctyl-4,9-dihydro-s-indaceno[1,2-b:5,6-b′]dithiophene-2,7-diyl)bis(3-octyl-[2,2′-bithiophene]-5′,5-diyl))bis(methaneylylidene))bis(5,6-dichloro-3-oxo-2,3-dihydro-1H-indene-2,1-diylidene)) dimalononitrile (IDT-BT-IC4Cl)

In a three-neck round bottom flask, to a stirring solution of compound **8** (0.3 g, 0.15 mmol) and 2-(5,6-dichloro-3-oxo-2,3-dihydro-1H-inden-1-ylidene)malononitrile (0.21 g, 0.78 mmol) in dried CHCl_3_ (30 mL), pyridine (1 mL) was added at ambient temperature, and the mixture was refluxed for 8 h. After water was poured, the mixture was extracted with CHCl_3_ and dried over MgSO_4_. After evaporating the solvent, the mixture was purified by silica gel column chromatography using methylene chloride:hexane (*v*:*v* = 3:1) as the eluent. Then, the compound was purified by recrystallization from CHCl_3_, and IDT-BT-IC4Cl was obtained as a black solid (0.24 g, 63.7%). ^1^H NMR (300 MHz, CD_2_Cl_2_, δ): 8.80 (s, 2H), 8.77 (s, 2H), 8.00 (s, 2H), 7.75 (s, 2H), 7.60–7.62 (d, *J* = 4.04 Hz, 2H), 7.50 (s, 2H), 7.28–7.30 (d, *J* = 3.98 Hz, 2H), 7.21–7.26 (t, 4H), 7.01 (s, 4H), 6.86–6.89 (m, 8H), 3.82–3.84 (d, *J* = 4.82 Hz, 8H), 2.87–2.93 (t, 4H), 2.62–2.68 (m, 4H), 1.72–1.79 (m, 8H), 1.09–1.52 (m, 80H), 0.87–0.97 (m, 36H), 0.68–0.71 (m, 4H). ^13^C NMR (500 MHz, CDCl_3_, δ): 185.81, 159.48, 158.60, 156.66, 156.01, 150.45, 148.72, 146.65, 143.35, 141.93, 141.53, 139.69, 139.52, 138.79, 138.00, 137.93, 136.24, 135.50, 134.61, 133.28, 129.68, 129.35, 126.93, 125.98, 125.34, 121.59, 121.14, 121.05, 116.24, 114.51, 114.40, 113.23, 71.24, 69.72, 64.34, 39.50, 31.99, 31.91, 30.71, 30.04, 29.66, 29.47, 29.38, 29.29, 29.26, 29.21, 24.10, 23.13, 22.76, 22.71, 14.18, 11.26. MS (MALDI-TOF) *m*/*z* Calcd: 2402.9886, Found: 2407.0256.

### 3.2. Fabrication of the OPV Devices

The newly synthesized IDIC-based NFA was used to fabricate an inverted organic photovoltaic device, which was composed of indium thin oxide (ITO), ZnO, PBDB-T:NFA, MoO_3_, and Ag layers (Figure 3). ITO and Ag were used as cathode and anode, respectively, and ZnO and MoO_3_ were used as electron and hole extraction layers. In inverted structure organic photovoltaic cells, the ZnO and MoO_3_ are typically used as electron and hole transport layer respectively. This is because, the ZnO and MoO_3_ have appropriate HOMO and LUMO levels with PBDB-T:NFA and they are very stable in moisture and oxygen environments [28,29]. ZnO was synthesized via the sol–gel method, which affords inorganic oxide powders. The precursor solution for the sol–gel synthesis of ZnO consisted of a mixture of 0.315 g of zinc acetate dehydrate, 10 mL of 2-methoxyethanol, and 0.158 mL of ethanolamine (Sigma-Aldrich, St. Louis, MO, USA), which was stored at room temperature for 24 h without heat treatment. PBDB-T was widely used as the photoactive layer as a donor material with NFA-based acceptor materials such as ITIC and IDIC. There are many previous research about PBDB-T: IDIC and ITIC bulk hetero-junction OPVs system [30,31]. In our research, the PBDB-T was purchased from 1-Materials Company (Dorval, QC, Canada). The photoactive layer was prepared by adding PBDB-T:IDIC-based NFA in 1 mL of 1,2-chlorobenzene at a weight ratio of 1:1, followed by heating at 80 °C for 10 h (20 mg/mL). To fabricate the solar cell, an ITO substrate (device area: 0.0225 cm^2^) was washed with acetone and isopropyl alcohol (IPA) for 10 min using an ultrasonic cleaner and then dried 100 °C for 5 h. After removing contaminants by ultrasonic cleaning, the substrate was made hydrophilic by UV-ozone treatment (UVC-30S) for 900 s, which facilitated the coating by lowering the contact angle between the solution and the substrate during spin coating [32]. After that, the ZnO precursor solution was spin-coated at 3000 rpm for 40 s at room temperature in an air atmosphere and then heated at 150 °C for 30 min to form a ZnO thin film (40 nm). Next, the PBDB-T:NFA active layer was spin-coated at 1000 rpm for 60 s in a glove box filled with N_2_ gas and then heated at 150 °C for 15 min to form a thin film (75 nm). On the coated device, 10 nm of MoO_3_ as a hole extraction layer and 100 nm of Ag as an anode were stacked at a vacuum degree of 3 × 10^−6^ using a thermal evaporator (SHIMADZU, Tokyo, Japan).

To evaluate the performance of the OPVs, IV characteristics were measured using a solar simulator (Xe55) that emits light of AM 1.5G (100 mW/cm^2^) and a Keithley 2400 source meter (Keithley Instruments, Solon, OH, USA), and EQE was measured with QuantX 300 (Newport Corporation, Irvine, CA, USA). Thin film analysis was performed to investigate the role of the NFA in the active layer. Absorption was taken with a UV–vis spectrophotometer (UV-2600i) (Light Machinery lnc, Ottawa, ON, Canada), and the roughness and shape of the surface were observed using an AFM (XE-100) (Park Systems, Suwon, Korea).

## 4. Conclusions

In summary, new bithiophene extended electron acceptors based on an *m*-alkoxyphenyl-substituted IDIC with three different end groups were developed, and their optical properties and thermal stability were analyzed. Then, inverted structure OPVs were fabricated using PBDB-T:IDIC-based NFAs and a BHJ system, and their characteristics were compared. The maximum absorption of NFAs was redshifted due to the presence of strong electronegative end groups. IDT-BT-IC, IDT-BT-IC4F, and IDT-BT-IC4Cl showed absorption maxima at 682, 710, and 726 nm, respectively. In particular, the absorption of IDT-BT-IC was almost 50 nm redshifted compared with that of ITIC-OEh without the bithiophene unit. According to a thermal stability analysis, the new NFAs based on ID-BT showed crystalline properties. The OPV devices showed the highest efficiency (3.37%) when IDT-BT-IC4F was used as an electron acceptor. The thin film of PBDB-T:IDT-BT-IC4F had a low average roughness of 0.263 nm, which facilitates the extraction of electrons and holes. Furthermore, the absorption above 800 nm led to the generation of more excitons, and as a result, a high J_sc_ of 8.31 mA/cm^2^ and PCE of 3.37% were achieved.

## Data Availability

All the data associated with present manuscript have been included in Appendix A.

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
