# Peer review of "New Bithiophene Extended IDIC-Based Non-Fullerene Acceptors and Organic Photovoltaics Thereof"

_molecules, 2022, doi:10.3390/molecules27031113_

Round 1

Reviewer 1 Report

Jeong et al. report new bithiophene extended IDIC-based NFAs and the inverted structure BHJ OPVs using PBDB-T:IDIC-based NFAs. One of the key claims is the A–D′–D–D′–A-structured IDIC derivatives with extended bithiophenes to increase the LUMO energy level for high open-circuit voltage (Voc). However, the reported Voc values are still in the range of 0.86V to 0.95V and the improvement in Voc is still marginal compared to the PV properties of best device incorporating NFA acceptors that is summarized in recent review paper (Table 2,  see Int. J. Mol. Sci. 2020, 21, 8085; doi:10.3390/ijms21218085). In addition, the reported efficiencies, particularly FF values, are too low (less than 50%), making it difficult to corelate potential impacts of different NFAs on PV performance. Refer to the reported efficiencies in Table 4 and Table 6 on the recent review paper (Int. J. Mol. Sci. 2020, 21, 8085; doi:10.3390/ijms21218085). The Reviewer believes that the OPV performance reported in the submitted manuscript needs to be significantly improved to make this work publishable.  For instance, the dark IV in Fig. 1(c) shows that there is almost no rectification between -1V and 1V, indicating that the fabricated OPV devices almost behave as a resistor, not a diode.   

Author Response

  1. Jeong et al. report new bithiophene extended IDIC-based NFAs and the inverted structure BHJ OPVs using PBDB-T:IDIC-based NFAs. One of the key claims is the AD′–DD′–A-structured IDIC derivatives with extended bithiophenes to increase the LUMO energy level for high open-circuit voltage (Voc). However, the reported Voc values are still in the range of 0.86V to 0.95V and the improvement in Voc is still marginal compared to the PV properties of best device incorporating NFA acceptors that is summarized in recent review paper (Table 2, see Int. J. Mol. Sci. 2020, 21, 8085; doi:10.3390/ijms21218085).

- (Author Response) : Thank for valuable comments. When designing the structure, high Voc was expected due to the improvement of the LUMO level, but it seems that the maximum Voc did not come out due to changes in other factors. Vloss is one of the most important factors limiting the PCE of the OPVs. In particular, the contribution from non-radiative Vloss should be reduced, which is mainly affected by the energetic interactions between the donor polymer and acceptors at the interfaces. Thus, the interfacial and morphological properties (i.e., domain size/purity, aggregation, and molecular orientation) of the BHJ blend should be optimized to decrease the non-radiative Vloss. We revised manuscript and added references

- (Revised Manuscript) Page 4, line 11-18: “When designing the structure, high Voc was expected due to the improvement of the LUMO level, but it seems that the maximum Voc did not come out due to changes in other factors. Vloss is one of the most important factors limiting the PCE of the OPVs. In particular, the contribution from non-radiative Vloss should be reduced, which is mainly affected by the energetic interactions between the donor polymer and acceptors at the interfaces. Thus, the interfacial and morphological properties (i.e., domain size/purity, aggregation, and molecular orientation) of the BHJ blend should be optimized to decrease the non-radiative Vloss [20]”

- (Add Reference) Page 10, line 48-50: “[20] Jun, Y; Tianyi, H; Pei, C; Yingping, Z; Huotian, Z; Jonathan, Lee, Y; Sheng-Yung, C; Zhenzhen, Z; Wenchao, H; Rui, W; Dong, M; Feng, G. & Yang, Y. Enabling low voltage losses and high photocurrent in fullerene-free organic photovolta-ics. Nat. Commun. 2019, 10, 570. https://doi.org/10.1038/s41467-019-08386-9”

  1. In addition, the reported efficiencies, particularly FF values, are too low (less than 50%), making it difficult to corelate potential impacts of different NFAs on PV performance. Refer to the reported efficiencies in Table 4 and Table 6 on the recent review paper (Int. J. Mol. Sci. 2020, 21, 8085; doi:10.3390/ijms21218085). The Reviewer believes that the OPV performance reported in the submitted manuscript needs to be significantly improved to make this work publishable.

- (Author Response) : We agreed with the reviewer’s comments. Our device efficiency is low, and FF is less than 50%. The newly synthesized NFA has interfacial and morphological issues with PBDB-T, and for this reason, the series resistance increases and the parallel resistance decreases, resulting in a low FF. The explanation related to the low FF has already been described in the page 4 line 16 (The fill factor (FF), which is the ratio of the actual maximum obtainable power to the product of Isc and Voc, ~ and IDT-BT-IC4F showed the highest PCE of 3.37% due to its high current density (8.31 mA/cm2) and FF (47%)) In addition, the influence of surface morphology on the effect of current density is explained on page 5 line 4 (The high Ra of IDT-BT-IC limits the movement of electron–hole pairs, resulting in the lowest current density value of 2.30 mA/cm2. ~ thereby improving the interface performance with MoO3 to achieve the highest current density of 8.31 mA/cm2). In our study, we newly synthesized NFAs and analyzed the performance of device with its morphological and interfacial properties. Therefore, although the device efficiency is low, we think that the novelty is sufficient and good results are obtained.

  1. the dark IV in Fig. 1(c) shows that there is almost no rectification between -1V and 1V, indicating that the fabricated OPV devices almost behave as a resistor, not a diode.

- (Author Response) : First manuscript, dark J-V in Fig. 1 (c) is log-scale. So, we revised Fig. 1 (c) and it is expressed linearly and added as an inset in Fig. 1(c). In a linear dark J-V, a large current appears at voltages above +0.5 V, which is typical of a diode characteristic

- (Revised Figure 1 (c)) Page 3, line 20: Figure 1. (c) dark J–V plot(inset: dark J-V linearly plot)

Reviewer 2 Report

  1. Please give the structure of donor, PBDB-T, and explain why select this material and how to obtain it.
  2. The PCE of the device is still low. I suggest the author to test other charge transfer materials to improve device performance. 

Author Response

  1. Please give the structure of donor, PBDB-T, and explain why select this material and how to obtain it.

- (Author Response) : Thank you very much for the reviewer's feedback. The PBDB-T was purchased from 1-Materials Company. In the reviewer's opinion, we added the structure of PBDB-T in the Figure 3. PBDB-T was widely used as the photoactive layer as a donor material with NFA-based acceptor materials such as ITIC and IDIC. There are many reference paper (ACS Appl. Mater. Interfaces. 2020, 12, 21, 24165-24173; Adv. Mater. 2018, 30, 8, 1705243) about PBDB-T: IDIC and ITIC bulk hetero-junction OPV system. We revised manuscript and added references

- (Revised Manuscript) Page 8, line 17-20: “PBDB-T was widely used as the photoactive layer as a donor material with NFA-based acceptor materials such as ITIC and IDIC. There are many previous research about PBDB-T: IDIC and ITIC bulk hetero-junction OPVs system [27,28]. In our research, the PBDB-T was purchased from 1-Materials Company”

- (Add Reference) Page 11, line 10-15: “[27] Li-Ming, W; Qingduan, L; Shengjian, L; Zhixiong, C; Yue-Peng, C; Xuechen, J; Haojie, L; Weiguang, X; Xiaozhi, Z and Tao, Z. Quantitative Determination of the Vertical Segregation and Molecular Ordering of PBDB-T/ITIC Blend Films with Solvent Additives. ACS Appl. Mater. Interfaces. 2020, 12, 24165-24173. https://doi.org/10.1021/acsami.0c02843; [28] Pei, C; Rui, W; Jingshuai, Z; Wenchao, H; Sheng-Yung, C; Lei, M; Pengyu, S; Hao-Wen, C; Meng, Q; Chenhui, Z; Xiao-wei, Z; Yang, Y. Ternary System with Controlled Structure: A New Strategy toward Efficient Organic Photovoltaics. Adv. Mater. 2018, 30, 1705243. https://doi.org/10.1002/adma.201705243”

- (Revised Figure 3) Page 8, line 4-5: Figure 3. (a) Device structure and (b) energy band diagram of inverted organic photovoltaics. (c) molecular structure of PBDB-T.

  1. The PCE of the device is still low. I suggest the author to test other charge transfer materials to improve device performance.

- (Author Response) : Thank you very much for the reviewer's feedback. In inverted structure organic solar cells, the ZnO and MoO3 are typically used as electron and hole transport layer respectively. The ZnO and MoO3 have appropriate HOMO and LUMO levels with PBDB-T : NFA and they are very stable in moisture and oxygen environments. We made a new OPV device with the structure reported in many papers (J. Mater. Chem. C. 2021, 9, 3901-3910; Electron. 2020, 87, 105944). The device structure is ITO/ZnO/PBDB-T : ITIC:MoO3/Ag, and it is the same as the device structure described in our manuscript. A high PCE of 8.78% was achieved with Jsc of 14.91mA/cm2, Voc of 0.89V, and FF of 66% as shown in below Figure. Therefore, the reason for the low efficiency of the solar cell is considered to be mainly due to the interfacial and morphological properties of PBDB-T : NFAs rather than the problem of the transport layer.

- (Revised Manuscript) Page 8, line 9-13: “In inverted structure organic photovoltaic cells, the ZnO and MoO3 are typically used as electron and hole transport layer respectively. This is because, the ZnO and MoO3 have appropriate HOMO and LUMO levels with PBDB-T:NFA and they are very stable in moisture and oxygen environments. [25,26].”

- (Add Reference) Page 11, line 5-9: “[25] Ioannis, I; Isaac, S; Giulia, L; Thomas, M. B; and Franco, C. Inverted organic photovoltaics with a solution-processed ZnO/MgO electron transport bilayer. J. Mater. Chem. C. 2021, 9, 3901-3910. DOI: 10.1039/d0tc04955g; [26] Bing, -Huang, J; Ping, -Hung, C; Yu, -Wei, S; Hsiang, -Lin, H; Ru, -Jong, J; Chih, -Ping, C. Surface properties of buffer layers affect the performance of PM6:Y6–based organic photovoltaics. Org. Electron. 2020, 87, 105944. https://doi.org/10.1016/j.orgel.2020.105944”

Reviewer 3 Report

This paper is of interest for the organic PV community and is well written. 

The chemistry was of great interest even if the photovoltaic performances were not very high. I recommend to accept it as it.

Author Response

Thank you for valuable comments. In the reviewer's opinion, the main of our paper is the newly synthesized IDIC material. And we have plans to further improve the performance through optimization of the OPV structure

Round 2

Reviewer 1 Report

Page 4: The series (Rs) and shunt resistance (Rsh) can be extracted by the diode curve fitting from the dark IV curves, not the conductivity.  Eq. 1 is valid for the resistor, not for the diode. In addition, there is not much correlation between the Jsc (or OPV performance) and the conductivity. For example, for thicker metal electrode, the conductivity may be low due to the additional sheet resistance but the Jsc won't be affected to much. Based on the dark IVs on Fig 1(c), all the OPV reported in the manuscript have large Rs (you can see Rs effect from the IV curve in the rage 4-5V. More importantly, very low EQE  of IC and IC4Cl (compared to IC4F), indicates that there are serious issues in device fabrication such as the usage of HTL and ETL that are not fully optimized. The OPV performance can be further improved to be at least decent by further optimizing OPV device fabrication processes, particularly MoO3 and ZnO.  

Author Response

  1. Page 4: The series (Rs) and shunt resistance (Rsh) can be extracted by the diode curve fitting from the dark IV curves, not the conductivity. Eq. 1 is valid for the resistor, not for the diode. In addition, there is not much correlation between the Jsc (or OPV performance) and the conductivity. For example, for thicker metal electrode, the conductivity may be low due to the additional sheet resistance but the Jsc won't be affected to much. Based on the dark IVs on Fig 1(c), all the OPV reported in the manuscript have large Rs (you can see Rs effect from the IV curve in the rage 4-5V.

- (Author Response) : We thank the reviewer for helping to publish a great paper. And we agree with the reviewer's feedback. As the reviewers commented, we found that the conductivity extraction with dark I-V was incorrect. Therefore, the content related to conductivity has been deleted in the first revised manuscript. As reviewer’s point out, the series resistance (Rs dark) extracted from dark I-V (4~5V range) has been added.

- (Revised Manuscript) Page 4, line13-14 : “The Rs and Rsh can be obtained from the J-V graph under light irradiation at 100 mW/cm2. Rs and Rsh are the inverses of the slopes at Voc and Jsc.”

- (Revised Manuscript) Page 4, line30-34 : “We extracted the series resistance in the 4-5V range of dark I-V in Fig. 1(c), and expressed it as Rs dark. The value of Rs dark is 1.35, 1.31, 40.27 Ω cm2 for IDT-BT-IC, IDT-BT-IC4F and IDT-BT-IC4Cl respectively. IDT-BT-IC4Cl has high series resistance at dark current, it shows a similar trend to Rs in the J-V graph under light irradiation at 100 mW/cm2.”

Table 1. Photovoltaic parameters of organic photovoltaic using different NFAs.

NFAs

Jsc (mA/cm2)

Voc (V)

FF (%)

PCE (%)

Rsh (Ω∙cm2)

Rs (Ω∙cm2)

Rs dark (Ω∙cm2)

IDT-BT-IC

2.30 ± 0.31

0.95 ± 0.03

45 ± 2

1.00 ± 0.20

830 ± 24

56 ± 8

1.35 ± 0.20

IDT-BT-IC4F

8.31 ± 0.51

0.86 ± 0.02

47 ± 3

3.37 ± 0.51

337 ± 15

18 ± 2

1.31 ± 0.15

IDT-BT-IC4Cl

3.00 ± 0.52

0.89 ± 0.03

29 ± 2

0.76 ± 0.24

361 ± 10

139 ± 9

40.27 ± 2.23

  1. More importantly, very low EQE of IC and IC4Cl (compared to IC4F), indicates that there are serious issues in device fabrication such as the usage of HTL and ETL that are not fully optimized. The OPV performance can be further improved to be at least decent by further optimizing OPV device fabrication processes, particularly MoO3 and ZnO.

- (Author Response) : Thank you very much for the reviewer's feedback. In inverted structure organic solar cells, the ZnO and MoO3 are typically used as electron and hole transport layer respectively. We have been studying optimization of ZnO and MoO3 for a long time. Also, we have published many OPV papers using ZnO and MoO3(Adv. Mater. Interfaces, 2020, 7, 2070057; Sol. RRL. 2021, 5, 2000673). Therefore, we fabricated thin films of ZnO and MoO3 by an optimized process in this manuscript. Recently, we fabricated an OPV based on PBDB-T:ITIC, the structure of which is ITO/ZnO/PBDB-T : ITIC:MoO3/Ag. A high PCE of 8.78% was achieved with Jsc of 14.91mA/cm2, Voc of 0.89V, and FF of 66% as shown in below Figure. Therefore, the reason for the low performance of the OPV is considered to be mainly due to the interfacial and morphological properties of PBDB-T : NFAs rather than the problem of the transport layer.

Reviewer 2 Report

Please add up-dated literature of nonfullerene acceptors in introduction section.

Author Response

Reviewer 2)

Please add up-dated literature of nonfullerene acceptors in introduction section

- (Author Response) : Thank you very much for the reviewer's feedback. We have added the recently reported papers of NFAs to the introduction.

- (Revised Manuscript) Page 1, line 45: “Representative nonfullerene acceptors are BTP-eC9 [10], BTP-4F-PC6 [11], Y6 [12], and IDIC [13].”

- (Add Reference) Page 10, line 22-33: “[10] Yunhao, C; Yun, L; Rui, W; Hongbo, W; Zhihao, C; Jie, Z; Zaifei, M; Xiaotao, H; Yong, Z; Chunfeng, Z; Fei, H and Yan-ming, S. A Well-Mixed Phase Formed by Two Compatible Non-Fullerene Acceptors Enables Ternary Organic Solar Cells with Efficiency over 18.6%. Adv. Mater. 2021, 33, 2101733. https://doi.org/10.1002/adma.202101733; [11] Jianquan, Z; Fujin, B; Indunil, A; Xiaoyun, X; Siwei, L; Chao, L; Gaoda, C; Han, Y; Yuzhong, C; Huawei, H; Zaifei, M; Harald, A; and He, Y. Alkyl-Chain Branching of Non-Fullerene Acceptors Flanking Conjugated Side Groups toward Highly Efficient Organic Solar Cells. Adv. Energy Mater. 2021, 11, 2102596. https://doi.org/10.1002/aenm.202102596; [12] Chao, L; Jiadong, Z; Jiali, S; Jinqiu, X; Huotian, Z; Xuning, Z; Jing, G; Lei, Z; Donghui, W; Guangchao, H; Jie, M; Yuan, Z; Zengqi, X; Yuanping, Yi; He, Y; Feng, G; Feng, L and Yanming, S. Non-fullerene acceptors with branched side chains and improved molecular packing to exceed 18% efficiency in organic solar cells. Nat. Energy. 2021, 6, 605-613. https://doi.org/10.1038/s41560-021-00820-x; [13] Song, Yi, P; Chiara, L; Joel, L; Yi,-Chun, C and Ji,-Seon, K. Organic Bilayer Photovoltaics for Efficient Indoor Light Harvesting. Adv. Energy Mater. 2021, 12, 2103237. https://doi.org/10.1002/aenm.202103237
